# Commentary: A Comparison of the Methods of the Pre-Slaughter Stunning of Cattle in Australia—Mechanical, Electrical, and Diathermic Syncope

**DOI:** 10.3390/ani14213141

**Published:** 2024-11-01

**Authors:** Gabrielle C. Musk, Craig B. Johnson

**Affiliations:** 1Animal Anaesthesia, Perth, WA 6107, Australia; 2School of Veterinary Science, Massey University, Palmerston North 4442, New Zealand; c.b.johnson@massey.ac.nz

**Keywords:** cattle, slaughter, stunning, animal welfare

## Abstract

Cattle in Australian abattoirs are slaughtered using a two-step method. The first step causes loss of consciousness by stunning, while the second step causes death from blood loss. The methods for causing unconsciousness include applying electrical current to the brain, applying mechanical force to the brain, or a new system of heating the brain tissue selectively (diathermic syncope). The features of different methods of stunning are reviewed to compare approval status in Australia, religious compliance, reliability, reversibility, time to effect, duration of effect, carcase damage, cardiovascular effects, logistics, and welfare implications. The pre-slaughter stunning of cattle with diathermic syncope is reversible, and does not cause carcase damage.

## 1. Introduction

Since cattle were domesticated as production animals, those involved with the husbandry, care, and preparation for human consumption of these animals have also taken responsibility for the timing and manner of their death. The method for slaughtering cattle has evolved from its original form where animals were physically restrained prior to the severance of a major blood vessel, such as the carotid artery, to a two-step process where animals are generally physically restrained to be stunned and then exsanguinated. Ethical perspectives, religion, and culture have guided the production, slaughter, and preparation of meat for human consumption, with halal (“lawful”) and kosher (“ritually pure”) methods still widely practiced by Muslims and Jews, respectively [1,2]. As appreciation of animal sentience and the importance of optimising animal welfare has developed, the methods used to slaughter animals, including cattle, have been evaluated and refined. These refinements have focused on the development of methods that ensure animal welfare is prioritised whilst meat quality is preserved within legislative, religious, and cultural contexts.

In Australia, the Industry Animal Welfare Standard for Livestock Processing Establishments reflects this prioritisation of animal welfare with the statement: “infrastructure and equipment shall not cause injury, pain, suffering or distress to the animal” (Section 5.2.1 in [3]). This standard encompasses activities associated with livestock receival at the establishment, livestock handling, restraint, stunning, and sticking procedures, which are all risks that potentially cause injury, pain, suffering, and/or distress [4]. While various strategies and practices mitigate these risks, it is essential that the establishment consistently fulfils these requirements through performance evaluation, internal audits, corrective actions, and regular review [3].

Regarding animal welfare at slaughter, the procedures involved with the transport of cattle from the farm of origin, the holding of animals prior to slaughter, restraint for stunning, effectiveness of the stun, and ultimate exsanguination must all be considered. The experience of an animal immediately prior to slaughter will be influenced by many factors including their interactions (with animals and people) during transport and at the processing plant, but the method of restraint and stunning may have a major impact on welfare at this time. This review focuses solely on the method of stunning, acknowledging that there are a multitude of other factors that contribute to the experience of an animal in this environment.

Extensive investigations have been undertaken to elucidate the impact of slaughter practices on both the welfare of cattle and the quality of meat obtained from those animals. While meat quality is a factor for consideration, it will not be discussed at length in this review.

The experiences of pain and fear contribute to the welfare state of animals whether they are caused by transport, holding, restraint, or otherwise [5]. While there are many welfare consequences of preparing cattle for slaughter, including heat stress, cold stress, fatigue, prolonged thirst, prolonged hunger, restriction of movements, lack of rest, and social stress, pain and fear during slaughter can and must be prevented [4]. Pain is defined as an unpleasant sensory and emotional experience associated with actual or potential tissue damage [6]. Animals must be conscious to experience pain, so it follows that if an animal has been rendered unconscious, pain is irrelevant at that point in time. However, if an animal were to regain consciousness during or after a painful procedure, pain may become a source of compromised welfare. In addition, if an animal was injured prior to stunning as a result of poor handling or restraint, or a mishap during the application of a stunning method, pain would be experienced and must be identified and treated immediately. When an animal is stunned and remains unconscious until death supervenes, there is no risk of any further experiences, including pain. Fear is a psychological concept that may be caused by unfamiliar environments, the disruption of social groups, handling, loud noises, odours, isolation, and pain [4,7]. While some of these experiences are difficult to avoid or control, such as fear, pain during death can be prevented by ensuring adequate stunning is performed.

The methods of slaughter can be categorised as (i) exsanguination without stunning, (ii) stunning prior to exsanguination, and (iii) post-exsanguination stunning [8]. Given that the Australian Animal Welfare Standards for cattle require that the killing method results in rapid loss of consciousness followed by death while unconscious, *stunning prior to exsanguination* is the method most relevant to this review [9]. In adult cattle, there are various methods of stunning, all requiring adequate restraint to ensure the accurate placement of a device to target the brain of the animal and create an unconscious state. For adult cattle, these methods include electrical stunning, mechanical stunning, and a novel dielectric (diathermic syncope) stunning system. The various attributes of these methods include reversibility, the time to onset of unconsciousness, effect on cardiac output and blood pressure, the duration of unconsciousness, required restraint, overall welfare impact, and necessary infrastructure. These factors will be discussed in detail in this review. It is important to note that, once an animal is effectively stunned, and therefore unconscious, the capacity for any experiences, including those associated with pain, are negated. As slaughter involves severing major blood vessels to cause death by exsanguination, an extensive incision through the sensitive tissues of the neck and/or thorax is required. This incision has been demonstrated to be painful if carried out while an animal remains conscious [10,11,12]. Therefore, the efficacy of a stunning method is vital to ensure that there can be no perception by the animal of the pain induced by cutting and subsequent bleeding, nor stress or discomfort due to shackling and hoisting. Timing becomes especially important in this part of the slaughter procedure, as the duration of time between the application of the stun and its induction of unconsciousness should be as short as possible [2]. Furthermore, the animal must not recover consciousness before exsanguination has progressed to a point where recovery is no longer possible.

The three categories of stun are mechanical (percussive), electrical, and diathermic syncope.

### 1.1. Mechanical (Percussive) Stunning

The aim of mechanical stunning is to cause significant concussion and trauma to the brain to interrupt its activities and create unconsciousness [13]. This injury to the brain can be caused by either a penetrating or a non-penetrating captive bolt. In both cases, the bolt gun must be held firmly against the surface of the forehead at a specific site to ensure adequate damage to the brain and the associated efficacy of the stun [14]. Because the placement and positioning of the projectile is critical, cattle must be restrained appropriately. A penetrating captive bolt will drive a rod or bolt through the bone of the skull into the brain tissue, while a non-penetrating captive bolt will eject a mallet-shaped knocker that leaves a depression in the skull at the site of application, causing damage to the underlying soft tissues and the brain [15]. In both instances, a secondary method of killing, usually exsanguination, must be performed [16]. This method of stunning (using either penetrating or non-penetrating captive bolt guns) is the most common method for stunning cattle in abattoirs in Australia [17,18].

The basic principles behind the operation of penetrating and non-penetrating captive bolts are the same and involve the transfer of kinetic energy from the moving bolt to the brain [19]; nevertheless, there are differences in the mechanism of action of the two different methods. Penetrating captive bolts produce a deleterious shockwave within and directly damage the brain, which is irreversible [20]. Because a relatively small area of the head is impacted by the bolt, highly focal kinetic energy damages the brain [17]. Non-penetrating captive bolts impart acceleration forces to the head when the bolt impacts the skull, and the forces applied to the head and brain have relatively low kinetic energy [17]. Most captive bolt guns in Australia are pneumatically powered, and the air pressure in the gun’s expansion chamber before shooting will affect the velocity of the bolt, the amount of kinetic energy delivered to the head and brain, and, consequently, the effectiveness of the shot [17]. The effectiveness of captive bolt stunning is influenced by the velocity of the captive bolt, the size of the animal, the positioning of the stunner, and the anatomy of the animal [21]. Various devices are commercially available and the cartridges must be selected according to the category or size of animal being slaughtered.

As previously mentioned, the use of a penetrating captive bolt gun immediately before the slaughter of cattle is intended to induce a deep and irreversible form of concussion or unconsciousness [20]. However, there is variation in the response to this method of stunning, and a ‘shallow depth’ of concussion may be observed when the animal fails to collapse and other signs indicate the need for a second shot to be administered [20]. These signs include the persistence of normal rhythmic breathing, eye movements such as nystagmus or rotation of the eyeball, and an intact corneal reflex [22].

When a non-penetrating captive bolt is used, the minimum air pressure is used to ensure an effective stun without damage to the skull. Skull damage is assessed in halal abattoirs to ensure that any damage is limited to indentation without the cracking of the bone [23]. Skull damage is graded from 1–6, with grades 3–6 involving cracking with or without displacement of bone and, potentially, the exposure of the brain tissue [23]. Only grades 1–2 are acceptable for halal. The main factors affecting the efficacy of a non-penetrating captive bolt stun are associated with the operator (skill and experience), the animal (breed, size, gender, age), and the facility and equipment (handling facilities and restraint box design have an impact on the presentation of the animal to facilitate the precise application of the equipment and ease of handling) [14].

The basic physics of mechanical stunning requires the delivery of a high amount of energy to the brain in as short a time as possible [24]. The kinetic energy from the captive bolt has to be transferred to the animal’s head and is proportional to the mass and the velocity of the object where kinetic energy = ½ × mass × velocity^2^ [24]. In accordance with this equation, the velocity of the bolt has a much greater influence on the kinetic energy delivered to the brain than the mass of the bolt.

### 1.2. Electrical Stunning

The electrical stunning of cattle can be achieved with a head-only method or head-to-body method. In both cases, electricity is used to induce unconsciousness and a current is passed through the brain to induce a grand mal seizure [16,24]. The signs associated with the effective induction of a seizure include the extension of the limbs, opisthotonos, the downward rotation of the eyeballs, tonic–clonic spasms, and eventual muscle flaccidity [16]. The head-to-body electrical stunning of cattle is irreversible and involves three sequential cycles: a head-only cycle to stun the animal; a cardiac cycle to induce ventricular fibrillation or cardiac arrest; and a discharge cycle to reduce convulsions after death [13]. The electrical characteristics to induce an effective and immediate stun followed by cardiac arrest in cattle have been investigated using EEG [13]. Cattle can be stunned effectively with currents as low as 0.46 amps for 3 s, but in the cardiac cycle, 1.51 amps is required to cause cardiac arrest [13]. The equipment for electrical stunning controls the voltage, frequency, waveform, and duration of the electric current [25]. Various factors affect the conduction of electricity and include skin thickness, fat covering, and skull thickness, which create resistance to conduction and introduce variability in response to the application of the current [25]. In all instances, it is important to note that, for animal welfare reasons, no current should pass through any part of the body before current has passed through the brain [24]. This approach ensures that the animal is unconscious before cardiac arrest is induced, and any pain that would be associated with cardiac arrest cannot be experienced [24].

Head-only electrical stunning is reversible, non-penetrating and is acceptable to some religious groups. However, the time to death in cattle by exsanguination must occur within the period of the electrical stun, so bleeding must be performed promptly to ensure that recovery does not occur before death due to exsanguination [25]. This period may be as short as 20 s or as long as 170 s [25,26].

The basic physics of electrical stunning is an implementation of Ohm’s law, where the relationship between the factors determining the flow of electricity are described [24]. The current that flows is measured in Amperes (A) and the driving force behind the flow of current is the voltage, measured in Volts (V). The ability of material to limit the flow of current is impedance and is measured in ohms (Ω). As current = voltage/resistance, to ensure adequate current flows through the brain, the voltage must be high enough to overcome the resistance inherent in the tissues of the animals. The current is either a pulsed direct current (DC) or an alternating current (AC). Pulsed DC is where electrons flow in one direction and the power source is periodically turned on and off to create the pulsed flow. An AC is when the electrons periodically change direction. Finally, the frequency of a current is the number of times the waveform is repeated in a second and is measured in Hertz (Hz) [24]. Water may be applied to the animal at the site of electrode placement to increase conductivity in some instances [13].

### 1.3. Diathermic Syncope Stunning

Diathermic syncope is a dielectric system for stunning cattle prior to slaughter. The mechanism of action is the selective increase of the temperature of the brain to the point that hyperthermic syncope occurs [18]. The effect on the animal is, however, more like an epileptic-type seizure with controlled tonic–clonic phases and unconsciousness lasting several minutes, as opposed to a faint [27]. The system avoids the potential for damage to the skull (associated with mechanical stunning) and the need for electro-immobilisation to protect personnel following electrical stunning. Furthermore, the duration of reversible unconsciousness is sufficient to allow adequate exsanguination and death [18]. Unconsciousness following the application of this method of stunning has been confirmed with EEG, which demonstrated seizure-like complexes, a pattern considered incompatible with awareness or consciousness [28,29,30].

Diathermic Syncope^®^ (DTS) has been trademarked as a method for the pre-slaughter stunning of cattle [31], and has the potential to address the requirements for halal and kosher markets as the stun is reversible; there is little or no damage to the carcase and electrical immobilisation is not required [18]. The settings for DTS are power (kW) and energy (kJ), with power influencing the time to onset of unconsciousness and energy influencing the duration of unconsciousness [18]. Notably, DTS may present a major advantage insofar as animals may be able to recover from unconsciousness, which is of particular interest to religious groups [28,32].

The physics of DTS is complex, involving the coupling of microwave energy and the target material for conversion to thermal energy. The ability of the material to store energy when a microwave field is applied is represented by the dielectric constant, and the ‘loss factor’ determines how readily the microwave energy is converted to heat within the material [29]. Ionic conduction and dipole rotation are the dominant mechanisms by which the majority of tissue materials store and convert energy to heat. When heating the brain, the dielectric properties of skin, fat, bone, and brain tissue must be taken into consideration, as they will contribute to the temperature, density, and moisture content of the material through which the microwave energy must pass [29]. Furthermore, as the tissue heats up, the dielectric loss factor increases and the hotter parts of the workload will be preferentially heated, creating further increases in the loss factor and non-uniform heating [29]. The DTS system was developed by ascertaining the dielectric properties of cattle tissues (being similar to humans) and predicting, then testing, the energy absorbed by the skin and bone of the bovine skull and the depth of penetration into brain tissue [29]. As the depth of penetration into the brain is the main area of concern with this method of reversible stunning, it was important to quantify the depth of penetration and its associated effects. The DTS system was developed to achieve ‘hyperthermic syncope’, which occurs when the brain temperature reaches 43 °C [29].

The reversibility of DTS has been demonstrated and described in detail (in a manuscript submitted for publication) [30]. The sequence of behavioural and EEG changes associated with stunning and recovery, including the time of initial physical response, eyelid movements, ear movements, nose twitching, convulsive movements, the return of spontaneous blinking, the return of corneal reflex, attempts to stand, awareness of surroundings, posture, and return to full alertness, were documented in six *Bos taurus* cattle [30]. These animals recovered completely within 30 min [30].

Exsanguination as the ultimate cause of death (following reversible stunning) is preferred for religious, cultural, meat quality, and food safety reasons. Death is caused by depriving the brain of the oxygen that is normally delivered in arterial blood. The anatomy of the blood supply to the brain in cattle means that the brain receives blood from both the carotid arteries via the anastomosing ramus of the maxillary arteries and branches of the vertebral arteries [33]. This anatomical structure means that it may take longer for the brain to be deprived of oxygen in cattle, compared to other species such as sheep, and it may, therefore, take longer to become incapable of maintaining awareness when a neck cut is performed [10,26]. In studies where cattle are slaughtered without prior stunning, awareness may commonly remain for as long as 85 s, with the longest duration of awareness recorded as 168 s [10,34]. Another study in adult cattle reported the loss of evoked responses in the brain for 20–126 s and the loss of spontaneous activity for 19–113 s after cutting in slaughter without prior stunning [35]. From these data, it may be inferred that, where animal welfare is a priority, stunning must be employed and must outlast the time it takes to die by exsanguination. To shorten the time to die from exsanguination, a thoracic stick may also be performed to hasten the onset of death [36]. Cattle present a particularly nuanced situation, with the vascular anatomy described above, as the risk of the prolonged perfusion of the brain (and associated brain function due to both collateral blood supply to the brain and the development of false aneurysms) is a potential animal welfare concern unless they are appropriately stunned [37].

In the context of processing healthy cattle, pain and any associated fear or distress are largely avoidable, and their minimisation must be a priority when examining methods of stunning. To mitigate the risk of pain and fear during stunning, various recommendations have been made: the appropriate restraint of the head and body are required for effective stunning; restraint should not itself induce pain and/or fear; firm restraint should only be instituted when the operator is ready to stun; stunning should only be performed when the operator is ready to bleed; irreversible stunning methods should be used or reversible methods must only be used when bleeding can be applied without delay; the efficacy of stunning must be checked and confirmed after stunning, prior to neck cutting, and during bleeding; and ineffective stunning must be identified immediately and addressed with a backup method [4].

The aim of this review was to examine the literature relevant to the stunning of cattle prior to slaughter; to compare existing methods of stunning with a novel system, focusing on ease of use, reliability, predictability, impact on welfare, alteration to the carcase, reversibility, and safety; and to identify opportunities for non-compliance with the Australian Industry Animal Welfare Standard [3] with regard to avoiding injury, pain, suffering, and/or distress. This review is warranted, as the adoption of diathermic syncope [30] may provide an opportunity to refine the process of pre-slaughter stunning in Australia.

## 2. Materials and Methods

Peer-reviewed publications, industry reports, government documents, and webpage information were considered for inclusion in the literature review of the attributes of the electrical, mechanical, and diathermic syncope methods of stunning. Targeted web searches (pubmed, google scholar, google, and the library databases of Murdoch University and the University of Western Australia) using search terms including slaughter, processing, stunning, percussive, non-penetrating captive bolt, electrical, microwave, diathermic syncope, and cattle were used to identify relevant documents between 1975 and 2024. Documents were included if they referred to any or all of the methods of stunning and their effect on the animal. One hundred and twelve documents were found and ninety-six of these were included. Documents were excluded if they were not considered relevant to cattle. The search parameters may have limited access to some research outputs, and in some cases, the publication was not related to the main focus of this review.

Ethical approval for this review was not required, as the existing literature was examined and no experimental work was conducted specifically for this publication.

## 3. Results

This review utilised 80 references: 56 peer-reviewed publications, 6 government documents, 10 industry publications, and 8 unpublished reports (2 doctoral theses, online information sheets, and statements) from 1975 to the present.

### 3.1. Approval Status in Australia

In accordance with the Industry Animal Welfare Standard [3], mechanical stunning methods (both penetrating and non-penetrating) are permitted with the caveat that the equipment be applied according to the manufacturer’s recommendations for appropriate position and power (charge, air-pressure etc) [3,9]. Electrical methods (both head-only and head-to-body) are also permitted on the condition that the electrode spans the brain or the brain and the heart, respectively, and that sufficient current to produce an immediate stun is applied [3]. Diathermic syncope is permitted in Australia in export meat processing facilities supervised by the Department of Agriculture, Fisheries, and Forestry (DAFF) as part of an Approved Arrangement, which is acquired upon application to DAFF and must be maintained in accordance with various requirements [38]. Methods such as controlled atmospheric stunning and low atmospheric stunning are not used in cattle.

### 3.2. Religious Compliance

The two major religions that have specific requirements for the handling of animals at slaughter are Islam and Judaism. Requirements relevant to slaughter are halal (Islam) and kosher (Judaism). Halal requires that an animal presenting for slaughter should be free from disease and that stunning should be non-fatal, reversible, and not impede blood loss [39]. Kosher requires that an animal be healthy and uninjured at the time of shechita and includes five Halachic requirements for the way in which the incision is made [40]. Both systems have additional requirements, but these are not considered to have any effect on animal welfare and so will not be considered further. Religious commandments relate more to the spiritual quality of meat rather than the inherent and perceived physical characteristics of meat such as tenderness, juiciness, and flavour. Therefore, if the minimum requirements for religious compliance are not met, meat may be considered spiritually worthless [1].

Both halal and kosher markets require that any method of stunning prior to slaughter must be approved by the relevant religious authority [41]. However, European Union regulation EC1099/2009 requires that all animals be stunned before slaughter [42]. There is a derogation that permits the slaughter of animals without stunning for religious consumption, but this is not exercised in all member states [41]. Scientists understand that many Muslim authorities allow the use of reversible stunning, and in some instances, the use of irreversible stunning methods, as long as the heart is still beating during exsanguination, while, at present, the Jewish community rejects stunning during shechita slaughter [41].

Non-penetrating captive bolts, head-only electric stunning, and the DTS system are considered to be appropriate for use during halal slaughter in Australia [29,39]. However, the interpretation of these criteria is not internationally consistent, and whether a method of slaughter satisfies a particular religious group is determined by the respective religious leaders in the region. There is at least one Australia-based Islamic organisation that has certified DTS as halal (personal communication with James Ralph). Mechanical and electrical stunning methods are not considered kosher [40], but further clarity is required to determine whether DTS may be suitable for kosher slaughter. A specific concern is whether the increase in brain temperature with DTS leads to permanent physical damage, rendering the animal non-kosher [27]. Recent unpublished evidence reports that thorough histological examination following DTS stunning has not revealed morphological changes in the brain tissue of cattle [30].

### 3.3. Reliability

In this context, ‘reliability’ is used to refer to the success rate of a single stun application to achieve unconsciousness. This outcome is important with whichever device is to be used. If the first stun fails, the risk of fear, distress, pain, additional tissue damage from repeated stuns, animal and personnel injury, and non-compliance with halal standards increases. Processing establishments must have procedures to document various activities including animal welfare monitoring, which encompasses the response to the application of the stunning device employed at that facility. All sites must have a backup plan to re-stun an animal promptly if the first method fails.

When considering the efficacy of a single application from a captive bolt gun, non-penetrating captive bolt guns are associated with a higher incidence of ineffective stunning and shorter duration of unconsciousness when compared with penetrative captive bolt stunning [4,14,17,39]. Factors that impact the reliability of non-penetrating captive bolts include the age of the animals as well as their skull structure and size, bone density and mineralisation, scalp thickness, mobility of the head and neck (related to restraint method), density and mass of neural tissue, and extent, nature, and direction of the concussive blow [39,43]. The need for two or more shots was more frequent for non-penetrating captive bolts compared to penetrating captive bolts (29% of cattle and 12% of cattle, respectively, required two or more shots to induce unconsciousness) [17]. The need for ‘repeat shots’ in cattle (including bulls, steers, and heifers) stunned with a penetrating captive bolt was 4.2%, despite 8.7% of these animals being ‘imperfectly stunned’ [20]. More recent publications describe penetrating captive bolt stunning in adult cattle, where stunning effectiveness was 99.4% [44] and 100% [45]. The improvement over time with this method is associated with refinements to equipment and the subsequent damage to the brainstem [45]. There are many studies reporting failure-to-stun rates with non-penetrative captive bolts and the results are variable, especially if adequate head and neck restraint is not in use (Table 1). Furthermore, the breed of cattle should be considered when interpreting the results, as certain breeds, e.g., Zebu or Brahman, may not be as well suited to the use of non-penetrating captive bolt stunning given the anatomy of the skull.

The efficacy of electrical stunning is influenced by various factors such as frequency [53], current [54], voltage [55], and the duration of application [56]. Electrical stunning has been associated with a 10% rate of re-stunning [57], although the authors of this study acknowledge that, in a controlled commercial setting, this figure may be lower. In a more recent study, the failure rate of electrical stunning was 1.6% in Australian abattoirs [14]. The application of water to the electrodes during the stun is used to reduce electrical resistance, especially in cattle with long hair, and this improves efficacy [58]. However, if too much water is applied, the current may be diverted over the surface of the animal and not through the brain, which will decrease the efficacy of this method [58]. The majority of published studies involve small stock, not cattle.

During development, DTS was reported to achieve a 97.1% effective stun with the first application [59]. Although the system has not been used on a large scale in a commercial setting, further refinements to the method have confirmed that the system is extremely reliable with a success rate of 97.8% (when n = 92) [30,60,61]. Given that compliance rates for halal criteria with non-penetrating captive bolt stunning vary from 75% to 92%, DTS data suggest that, when DTS is used, there will be fewer non-compliant carcases than when non-penetrating captive bolts are used [61]. The failure of DTS is associated with poor waveguide positioning, which is a focus of future refinement for this method to maintain a low failure rate [30].

Despite the published data (Table 1), when penetrating or non-penetrating captive bolt devices are used, a 96% effective stun with the first application rate is considered acceptable by the Industry Animal Welfare Standard [3]. When electrical stunning is used, at least 99% of animals must have the electrodes applied in the optimum position, and 98% or more of the animals must not show any sign of starting the process of return of consciousness [3]. An acceptable stun rate for DTS has not been defined by the Australian Meat Industry Council in the Industry Animal Welfare Standard for Livestock Processing Establishments Preparing Meat for Human Consumption.

### 3.4. Reversibility of Unconsciousness

Whether an animal can regain consciousness after stunning is an important criterion for religious compliance. Reversibility and duration of effect are interlinked, and the latter is discussed below. Carcases are considered non-compliant with halal requirements if skull damage (e.g., following non-penetrating captive bolt stunning) is likely to be associated with permanent neurological damage and an inability to recover [23].

Penetrating captive bolts cause physical damage to the brain and are generally considered to be associated with irreversible unconsciousness [24]. However, non-penetrating captive bolts can cause the transient elimination of sensibility, i.e., reversible unconsciousness [24]. Head-only electrical stunning causes a seizure from which the animal may recover, while an electrical current flowing from the head to the brisket or back or front leg will also pass through the heart and can cause cardiac arrest (death) from ventricular fibrillation, which is irreversible [24]. Diathermic syncope can also induce reversible unconsciousness [30,62]. However, with DTS, if the energy delivered is greater than 300 kJ, insensibility may progress to death [18].

### 3.5. Time to Effect

The time between the application of the stun and unconsciousness is important as, during this time, restraint must be maintained and any fear and distress must be minimised. It is also important that the time to effect is predictable and consistent to ensure a regular flow of animals for processing.

With penetrating captive bolts, the time to effect is not defined in otherwise comprehensive studies of efficacy [20]. For electric stunning, a current of 1.2 A must pass through the brain for at least 2 s, but the time to achieve unconsciousness is unclear [58]. When applied for 3 s, currents of >0.46 A at 50 Hz induced epileptiform activity in the brain (consistent with unconsciousness) in cattle undergoing head-only electrical stunning [13].

Following the application of DTS, with energy applications between 200 kJ and 360 kJ, the loss of posture occurred between 1 and 8 s after the onset of energy delivery [18]. In animals where the stun was not effective, the ‘leakage’ of energy into the environment, rather than the penetration of the brain, was due to the interruption of the contact between the waveguide and the forehead [18]. The current recommended commercial application of DTS uses 160–200 kJ of energy at 18kW to cause a tonic–clonic response in less than 5 s [30].

### 3.6. Duration of Effect

For all methods of stunning, the duration of unconsciousness is a vital detail, as the period of unconsciousness induced by the stun must be longer than the time taken to cut the neck and for exsanguination to render the animal no longer capable of achieving awareness.

A successful penetrating captive bolt or head–body electrical stun will result in the loss of consciousness that will continue until death supervenes [63] due to the cessation of breathing in the case of the penetrating captive bolt or the cessation of circulation in the case of head–body electrical stunning. Nevertheless, with penetrating captive bolts, it is best to cut the neck within two minutes of stunning to preclude the risk of recovery from the stun prior to death [64]. With head-only electrical stunning, some animals may regain consciousness prior to the completion of exsanguination [25,55,65]. The head-only electrical stunning of calves lasts 34 s [66], and for cattle, the duration of the tonic and clonic phases is 5–10 s and 10–60 s, respectively [25]. The time to the return of rhythmic breathing in cattle following a head-only electrical stun was on average 50 s [13]. As it may take 55 s for death by exsanguination (when both carotid arteries and both jugular veins are severed), these techniques require a short stun-to-bleed interval [25]. To ensure rapid death by exsanguination, a thoracic stick is often also performed to cause death within 30 s [36]. Consequently, the Humane Slaughter Association mandates that animals must be bled within 15 s of stunning to ensure that death supervenes during the duration of the stun [25]. The duration of effect of a non-penetrating captive bolt is not comprehensively described, but has been reported to be ‘any time up to 5 mins’ in cattle [63]. In sheep and calves, recovery from unconsciousness did not occur in less than 2 min [47], and in calves, the absence of righting behaviour lasted for at least 60 s [48]. The duration of effect, as evidenced by the absence of corneal reflex and EEG data, of DTS is reported as between 100 and 170 s in one study [18], and between 152 and 369 s with the recommended commercial parameters [30].

When penetrating captive bolt stunning, non-penetrating captive bolt stunning, and post-neck-cut stunning were compared, the data showed that there was no difference in the time to the cessation of visible vital signs and reflexes, i.e., death [67]. The average time from post-slaughter to death (confirmed by EEG) was 190 s in this study of 40 heifers and steers [67].

### 3.7. Carcase Damage

Carcase damage must be minimal and within allowable and defined ranges to comply with religious requirements. For halal requirements, the indentation of the skull without cracking (or displacement) is the maximum allowable damage to the carcase [23,68].

The application of mechanical energy from a non-penetrating captive bolt gun may fracture the skull [19,39,69]. In one study, 60% of heads were examined after stunning with a non-penetrative device, and profound injuries to the frontal bone in the impact area of the bolt, including inner and outer bone laminae and part of the dura mater, were evident [8]. Sub-dural haemorrhage has also been observed following the application of non-penetrating captive bolts [63]. Haemorrhage to varying extent beneath the impact site of the bolt and around the brain stem has been observed in multiple studies [8,47,48,49,69]. Penetrating captive bolts cannot be used without causing damage to the skull [15].

Head-only electrical stunning does not cause physical damage, but carcase quality issues such as ecchymoses (‘blood splash’) can adversely affect meat aesthetics [55,70]. A comparison of carcase quality, including damage such as superficial bruising, was performed on carcases stunned with either a penetrating captive bolt or the DTS system [71]. However, this study focused on carcase quality in the context of commerciality, as opposed to carcase damage and its influence on religious criteria. No visible damage to the brain or carcase was observed following DTS stunning [18]. The most recent study reporting the use of DTS reported that carcase damage did not occur, with the exception of the depilation of the forehead at the application point in 4 of 92 animals [30].

### 3.8. Cardiovascular Effects

Efficient and rapid bleeding is essential in the slaughter process to ensure that death by exsanguination occurs before recovery from the stun. Rapid blood loss will cause a dramatic drop in blood pressure leading to shock, failure of physiological compensation mechanisms, and death [72]. The circulating blood volume of animals is 7–8% of body weight [73,74], and a 30–40% loss of blood volume is associated with loss of consciousness [75]. Exsanguination is important for religious reasons, and the quality of meat is also better when more blood is lost during slaughter [36]. In addition, for hygiene, food safety, and shelf-life reasons, as much blood as possible should be expelled from the carcase [55,76]. The most common methods for bleeding include transverse neck sticking (severance of both carotid arteries and jugular veins) and thoracic/chest sticking (cutting the skin longitudinally from the neck down to the chest along the midline and then cutting into the chest near the heart to sever the aorta and vena cava) [55]. Various factors influence the rate and efficiency of bleeding: cardiac arrest at stunning (e.g., during head-to-body electrical stunning); the size and number of blood vessels severed; vasoconstriction or vasodilation in the capillary bed; the size of the sticking wound; the position of the carcase (horizontal or vertical); clonic activity causing the movement of blood towards the sticking wound; and tonic muscle contractions squeezing blood capillaries and vessels [55]. Studies on the effect of the slaughter method on blood loss have yielded contrasting results in various species. In general terms, stunning methods that cause cardiac arrest (e.g., head-to-body electrical stunning) slow blood loss [55]. Muscle spasms and the associated body movements that occur after stunning may aid the removal of blood from the muscles and major internal organs, while stress immediately prior to stunning may increase the release of catecholamines, which cause vasoconstriction, promoting the removal of blood from muscles [55]. A potential issue with electrical stunning is that low electrical frequency results in poor bleeding and high incidences of carcase defects [55].

If bleeding time needs to be shortened, thoracic sticking, as opposed to neck cutting, may be performed to increase the rate of blood loss [36,77]. In addition, electrical stimulation after stunning may further increase blood loss and the efficiency of bleeding [77]. However, this approach must not be taken within 30 s of sticking [42].

The bleeding time following non-penetrating captive bolt stunning has been reported to be 304 ± 70 (mean ± standard deviation) seconds [78].

With DTS, there is no requirement to perform a backup thoracic stick or electrical immobilisation, as blood flow is strong and exsanguination is rapid [60]. Blood flow is noted to be visibly strong and the blood pulses out with each heartbeat, especially in the early stages of bleeding out [60].

Ecchymotic haemorrhage in muscles (or ‘blood splash’) occurs following non-penetrative percussive stunning and electrical stunning [55,61]. The mechanism of ecchymoses is not entirely understood and may be caused by a transient increase in blood pressure and the rupture of small blood vessels in the muscles because of an increase in cardiac output, an increase in vascular resistance (vasoconstriction), or both [79]. The effect of the stunning method on the severity of blood splash appears to be no stun < non-penetrating captive bolt < penetrating captive bolt < head-to-body electrical < head-only electrical [1]. To date, no evidence of blood splash or ecchymosis has been observed with DTS [60].

### 3.9. Logistics

#### 3.9.1. Restraint

For all methods of stunning, precision is required to ensure the target region of the head can be accurately identified to ensure that the brain is directly affected. In all cases, once positioned, the animal should be stunned without delay. The method of restraint will vary, but must be designed and operated effectively to allow the animal to be positioned for effective stunning [3]. In all instances, the stunning box should be narrow enough to prevent the animal from turning around and the floor should be non-slip so the animal can stand without losing its footing [80]. For electrical stunning and for the application of a non-penetrating captive bolt, a head holder should be used to optimise the positioning of the stunning device [80]. The Humane Slaughter Association suggests that, for the electrical stunning of cattle, the restraint system should include a neck yoke, chin lift, and rump pusher to facilitate accurate electrode placement [25]. For DTS stunning, the head must be secured to enable the placement of the waveguide that emits the energy onto the forehead. If contact between the waveguide and the forehead is compromised, inconsistent delivery of energy to the brain may influence the reliability of the technique [18]. The specific challenges of restraint for DTS stunning have been investigated and refined to ensure that the reliability of the technique is high [18]. In brief, the animals must be individually restrained in a stun box with neck yoke, head capture system, and chin lift—all within a Faraday cage to comply with occupational health and safety requirements [18]. The system has been set up to interrupt the delivery of energy if contact with the forehead is inadequate; however, the equipment provides real-time feedback that allows the operator to immediately address the positioning and instantaneously re-stun the animal, if necessary [18].

#### 3.9.2. Equipment Maintenance

Non-penetrating captive bolt guns should be cleaned regularly to remove the accumulation of silica or carbon, which reduces the power of subsequent shots [39]. Penetrating captive bolt guns have rubber rings that are fitted around the bolt to return the bolt to the breach after maximum extension. These rings should be replaced every 4000 shots [64]. Otherwise, there is a danger that the bolts will eventually shear and be dangerous to staff [64]. Furthermore, if this equipment is used repeatedly in a session, the performance will be reduced, so the bolt and barrel should be checked regularly [81].

Head-only electrical stunning equipment is generally comprised of either scissor or fork tongs and a control device with or without a monitor [25]. Electrodes must be cleaned regularly every 20–25 animals with a wire brush or powered wire wheel to ensure there is minimal contact resistance [25]. The control box and tongs should be stored in a dry environment and, in between stunning procedures, the tongs should be seated on a mounted wall bracket or in a cleaning station [25]. The Jarvis Beef Stunner was developed for adult cattle to induce a stun, cardiac arrest, and spinal discharge (three consecutive cycles) [82].

DTS requires a rotating box or standing crush, Faraday cage and associated interlocks, the DTS stunning system with waveguide and control box, and a landing table or cradle for exsanguination [61]. There are daily, weekly, and monthly care requirements for the DTS equipment, which cover the generator, the applicator, and the restraint unit [60]. The waveguide does not need regular calibration, although the waveguide, applicator, and Faraday cage should be inspected each day for damage and condition [60].

#### 3.9.3. Throughput

Processing rates for different stunning methods are not readily available in the literature. The time taken to process an animal will depend on pre- and post-stun handling strategies and the time to address unexpected events such as failed stuns or other mishaps. A monorail system that delivers cattle to the restraint area is likely to decrease pre-stun handling times when compared to ‘walk-in’ methods, whereas post-stun positioning for exsanguination and the need for electrical immobilisation impact how many animals can be moved through the system. If bleeding can be performed on the carcase away from the landing area, the turnover is greater, but if bleeding is performed in the cradle, the next animal cannot be stunned until the bleeding of the previous animal is complete.

The attributes of DTS that will optimise efficiency include the reliability and elimination of any time lost to re-stunning [61], fast bleeding, and the availability of a rotating box (allowing for a second animal to be loaded while the first is exsanguinated). These facilities have a processing capacity of two animals per minute (i.e., 120 per hour) [60].

#### 3.9.4. Need for Electrical Stimulation of the Carcase

The electrical stimulation of carcasses is standard practice in beef processing to accelerate the tenderisation of meat by rapidly lowering the pH, and preventing cold shortening (very tough meat) [83]. However, the effects of electrical stimulation on meat quality are not understood to the extent that they can be quantitatively described. Furthermore, the interaction of electrical stimulation with other electrical inputs such as electrical stunning or electrical immobilisation requires further investigation [83]. The practice of electro-immobilisation after reversible stunning and prior to exsanguination is not acceptable and is not endorsed by the Australian Quarantine and Inspection Service [63]. However, electro-immobilisation after exsanguination is confirmed to have caused irreversible insensibility and is acceptable in Australia and Europe [42,63]. There are, however, exceptions, and with head-only electrical stunning, electric immobilisation is routinely used to improve operator safety. As the duration of the application of the electrical stun will influence the duration of unconsciousness (the longer the duration of application, the less risk there is to operators when restraining, lifting, and slaughtering animals), electro-immobilisation is not always necessary [56]. Animals stunned with DTS do not require electro-immobilisation [18].

#### 3.9.5. Safety

Handling, restraining, and, ultimately, rendering cattle unconscious present various hazards. With all stunning methods, the hazards to both animals and personnel can be mitigated by ensuring that operators are appropriately trained, skilled, and experienced [63]. In addition, risks can be managed by ensuring facilities are appropriately designed and regularly monitored and maintained.

There are various challenges when using a captive bolt gun: operator safety associated with the recoil of a captive bolt; the restraint of the head within reach of the operator; and the proper positioning of the shot in agitated animals [19]. Furthermore, electrical immobilisation post stunning is often required to prevent body movement and danger to personnel. This is only permitted 30 s after neck cutting so that signs of impending recovery of consciousness are not suppressed during the critical stun-stick interval [63].

All electrical stunning equipment is potentially dangerous to staff. Equipment should be checked regularly and maintained by a qualified electrician to ensure that current always flows preferentially between the two electrodes in an isolated circuit [25]. The switch or trigger on the control unit should only be pushed or turned on when current is needed (i.e., do not permanently activate the unit or the electrodes will be permanently ‘live’). A preset timer, which regulates the duration of the current flow, is also useful [25]. The control box must always be kept somewhere dry and protected from physical injury, and a warning light system will display when the equipment is being used to deliver a current [25]. Wet conditions are potentially dangerous, as water conducts electricity. Electrical immobilisation is often required with head-only electrical stunning, but is only permitted 30 s after neck cutting [63].

Microwave safety is relevant to the DTS stunning method and is addressed by the design of the Faraday cage, which prevents any microwaves from escaping into the surrounding environment [60]. If the cage or choke fails, a radiation interlock system will shut down the DTS system and prevent further operation if an unsafe level of radiation is detected [60]. Animals stunned with DTS cannot make violent movements that endanger workers due to the design of the restraint apparatus, so electrical immobilisation is not required.

The components of this review are tabulated for each of the methods of stunning discussed herein: mechanical stunning, electrical stunning, and diathermic syncope stunning (DTS) (Table 2).

### 3.10. Welfare Impact

There are multiple opportunities to optimise animal welfare in a processing facility. As previously mentioned, the steps from receival, holding, and presentation for stunning prior to slaughter may all impact the welfare of animals in a neutral, positive, or a negative way. While the intention of the Industry Animal Welfare Standard is to avoid injury, pain, suffering, or distress, each of these steps must be carefully managed to be compliant [3]. The focus of this review is on pre-slaughter stunning, where the key to optimising animal welfare at the point of slaughter is to ensure that animals are unconscious at the time of exsanguination so they do not feel pain associated with the cutting of the skin. All efforts should be made, in accordance with expectations, to ensure “the minimisation of the risk of injury, pain and suffering and the least practical disturbance to animals” [85]. If stunning is ineffective in the first instance, there is a risk to animal welfare from being subject to extended restraint, partial stunning, the repeated application of a stunning method, shackling and hoisting while conscious, and partial or complete consciousness during exsanguination [14].

The evaluation of blood parameters and the electroencephalogram to reflect stress during slaughter without stunning or with penetrating captive bolts or non-penetrating captive bolts revealed that no stunning is most stressful, and that penetrating captive bolts were less stressful than non-penetrating captive bolts [84].

Defining various acceptable timeframes for the stages of slaughter will provide a framework for compliance whereby the time for the induction of unconsciousness due to successful stunning, the ‘stun to stick’ interval, and the time for death from exsanguination ensures that consciousness is not recovered at any point. Until these frameworks exist, the welfare impact of pre-slaughter stunning methods cannot be directly compared. Nevertheless, the best welfare outcomes come from methods of slaughter that cause death after the induction of unconsciousness without any risk of recovery during the process.

## 4. Discussion

The aim of this review was to compare a novel pre-slaughter stunning method (DTS) to established percussive and electrical methods in cattle. Stunning prior to exsanguination at slaughter may be reversible or irreversible (permanent and causing death). In this review, penetrating captive bolt (percussive or mechanical) stunning and head-to-body electrical stunning are both irreversible methods of stunning that, although performed in Australia, do not meet the requirements for religious and cultural slaughter, where animals must be able to recover from the method of stunning. Nevertheless, these methods were included, as they are an important feature of the evolution of stunning methods and the quest for a method that satisfies the various criteria for acceptable slaughter, especially religious and cultural requirements, Animal Welfare Standards, and meat quality. While DTS is a reversible method of stunning, the most appropriate comparisons for the Australian context are with non-penetrating captive bolts and head-only electrical stunning methods.

To identify the best method of pre-slaughter stunning, side-by-side studies in a controlled environment will enable the comparison of parameters of interest, such as reliability, reversibility, time to effect, duration of effect, carcase damage, and logistics. As these studies do not exist for DTS a review of existing data, accepting different contexts, different cohorts of animals, and different experimental outcomes is the only approach that can be taken at the current time. DTS has yet to be utilised in a commercial environment, but the data support the premise that DTS is reliable and reversible, allows time for exsanguination, does not damage the carcase, and can be performed with the appropriate infrastructure and staff training.

A key component of ensuring minimal distress during slaughter is effective stunning. Stunning is performed to induce unconsciousness, which is defined as the loss of awareness occurring when the brain’s ability to integrate information is blocked or disrupted [16]. In animals, unconsciousness is often confirmed by loss of the righting reflex [16]. While this integrated whole-animal response is easily observable, it can be difficult to assess in restrained animals, especially cattle. Furthermore, the loss of a righting reflex can also occur when animals are conscious. The measurement of electrical activity in the brain can also be used to evaluate (un)consciousness. Although this approach requires sophisticated equipment and the complex analysis of the electroencephalogram (EEG), it is an extremely useful research tool [86,87]. There is, however, speculation on whether the EEG can be used to confirm the precise time of onset of unconsciousness [16]. Despite this concern, the EEG remains a useful method to assess whether an animal is unconscious. Electroencephalographic data can be analysed statistically and, when used in combination with observations of physical behaviours, the evaluation of the induction of unconsciousness and recovery is enhanced [88]. When developing a novel method of stunning, EEG data are a valuable means of assessing the ability of a technique to produce unconsciousness. Once a method is confirmed to be suitable for the induction of unconsciousness, physical indicators of unconsciousness may be relied upon in the processing plant on a day-to-day basis, as the EEG is not a crush-side tool.

In cattle, various observations can be made to increase confidence in the effectiveness of stunning and confirm unconsciousness in real time [89,90]. These observations include, as previously mentioned, the loss of the righting reflex, but also the absence of corneal and palpebral reflexes, ability to track moving objects, extension of the limbs, opisthotonos (backward arching of the head and neck), downward rotation of the eyeballs, tonic (stiffened muscles) spasm changing to clonic (twitching or jerking muscles) spasm, and eventual muscle flaccidity [16,20,91]. Signs of inadequate stunning include a stiff curled tongue, the rhythmic puffing of the cheeks, twitching of the nose, and retraction of the tongue [91].

Once the animal is unconscious and has been killed, it is vital that death is confirmed before the carcase progresses through to processing. A combination of observations is the most reliable approach to confirming death: the absence of a pulse, no breathing, no corneal reflex, no response to a painful stimulus, greying of the mucous membranes (gums and tongue), and rigor mortis. The latter is the only sign that, in isolation, confirms death, but does take time to set in. The other observations must be made collectively to be confident in the assessment of death. Monitoring should continue during neck cutting and during bleeding [22]. Detailed descriptions of the physical variables that indicate unconsciousness and death, and their feasibility, sensitivity, and specificity, are available in a 2013 publication from the European Food Safety Authority [22]. In brief, indicators of unconsciousness include posture (immediate collapse without efforts to regain posture); breathing (immediate absence of breathing); tonic seizure (onset after collapse); muscle tone (relaxed legs, floppy ears and tail, relaxed jaw, and protrusion of the tongue); response to nose prick or ear pinch (negative response); body movements in response to sticking (not intentional or purposeful); vocalisation (no purposeful grunting, bellowing, or mooing); eye movements (fixed eyes); palpebral reflex (negative); corneal reflex (negative); blinking (negative); and pupillary reflex (negative) [22].

Another major factor contributing to the utility of stunning methods is the end product and its quality. Meat quality is generally described in terms of aesthetic, tactile, masticatory, functional, nutritional, health, convenience, and environmental impact attributes [1]. Each of these features of meat quality can be affected by the methods employed before, during, and after slaughter [19,92]. Extensive research has been performed to dissect the nuances of the impact of a procedure on meat quality at all stages (before, during, and after slaughter), and how to assess their impact physiologically, biologically, and, ultimately, by taste [19,93,94]. In Australia, a beef grading system was introduced in 1998 to regulate the quality and consistency of meat palatability [94,95]. This system, known as Meat Standards Australia (MSA), enables the prediction of intrinsic eating quality based on large-scale consumer sensory testing to determine the impact of management, carcass phenotype measures, and post-mortem processing on palatability [93,95]. Given that consumers will pay more for higher quality meat when they have confidence in the product, the MSA system has enabled the consistent production of high-quality meat for both domestic and international markets. Fatigue and stress prior to slaughter can result in depleted glycogen (energy) stores in muscle, while acute stress can activate the ‘flight-or-fight’ response, increasing metabolic rate. Depleted glycogen can result in a higher ultimate pH in meat, while an increased metabolic rate can result in an abnormally rapid fall in pH during the rigor mortis process. Both scenarios can adversely affect meat quality and can be somewhat avoided by ensuring that stress prior to stunning is minimised [92,93].

## 5. Conclusions

For each method of pre-slaughter stunning, there are specific techniques for restraint and optimal settings to ensure the delivery of kinetic energy, electrical current, or electromagnetic energy to cause changes in the brain that create unconsciousness. Indeed, if applied successfully, the methods of stunning discussed in this review will ensure that cattle do not experience pain and distress during slaughter. However, when identifying a method that has acceptable reliability time for death by exsanguination before recovery, and hence reversibility, non-penetrating captive bolts are superior to penetrating captive bolts and electrical methods of stunning. As these methods have been utilised for considerably longer than DTS, familiarity with the processes of stunning using these methods is much greater than with DTS. However, DTS has been developed in a methodical and systematic way, with particular attention to creating a method that optimises animal welfare, and reliably and reversibly stuns cattle without damaging the carcase [18,27,29,59]. Notwithstanding the various barriers to the adoption of a new method of stunning, DTS appears to be a method that addresses the animal welfare concerns of international and Australian beef processing industries [61].

## Figures and Tables

**Table 1 animals-14-03141-t001:** Reported failure rates of non-penetrative captive bolts.

Reference	Species	Results	Relevant Details
Oliveira et al., 2018 [17] Meat science Peer reviewed	Zebu and Zebu-cross cattle *n* = 92 non-penetrating captive bolt *n* = 363 penetrating captive bolt	Overall, 29% required two or more shots with non-penetrating captive bolt Overall, 12% required two or more shots with penetrating captive bolt	Based on physical observations, they concluded penetrating captive bolts were more effective at ensuring unconsciousness in cattle than non-penetrating captive bolts
Gibson et al., 2019 [46] Meat science Peer reviewed	Zebu cross bulls *n* = 11 non-penetrating captive bolt *n* = 20 penetrating captive bolt	Overall, 82% showed EEG signs of complete unconsciousness with non-penetrating captive bolt, while the rest required a second shot Overall, 100% showed EEG signs of complete unconsciousness with penetrating captive bolt	Based on EEG assessment, they concluded non-penetrating captive bolts were less effective at inducing unconsciousness than penetrating captive bolts
Hewitt 2016 [14] Report by Australian Meat Processor Corporation Not peer reviewed	Various cattle captured in audit reports in 2014/2015 in 42 facilities In total, 3235 cattle assessed during audit process	Ineffective stun from non-penetrating captive bolts in 3.6% of animals Ineffective stun from penetrating captive bolts in 0.4% of animals Ineffective stun from electrical stun in 1.6% of animals	The main factors affecting efficacy of non-penetrating captive bolts were associated with the operator (skill), the animals (anatomy), the facility, and the equipment
Blackmore 1979 [47] Veterinary Record Peer reviewed	Lambs, adult sheep, and calves	There was a 20% failure rate with non-penetrating captive bolts	Brain haemorrhages frequently occurred in calves
Lambooy et al., 1981 [48] Conference proceedings	Veal calves	There was a 21% failure rate with non-penetrating captive bolts	Limited details available
Hoffman 2003 [49] Doctoral thesis	Cattle *n* = 1248 in two large processing facilities	In total, 12% of animals had to be stunned twice	Although non-penetrating captive bolts protect personnel from exposure to potentially infected brain material, the animal welfare results are unsatisfactory
Endres 2005 [50] Doctoral thesis	Cattle *n* = 5552 in large processing facility	Highest re-stun rate with bullocks There was a 16.7% failure rate overall	Comparison of two different non-penetrating captive bolt devices (Jarvis and EFA) Frontal bone injury and haemorrhage around the brain was observed in 60% of heads Non-penetrating captive bolts are less suited to routine slaughter than penetrating captive bolts
Collins et al., 2020 [51] Animal welfare Peer reviewed	Holstein calves 4–5 months *n* = 12	Overall, 100% success with penetrating and non-penetrating captive bolts in sedated calves	An occipital approach was studied in this cohort of sedated calves
Supratikno et al. [52] Acta Veterinaria Indonesiana	Adult Australian Brahman-cross cattle in Malaysia *n* = 460	Stunning success 74.35%	Low success influenced by location of shot, shooting placement distance, and anatomy of the skull

**Table 2 animals-14-03141-t002:** Summary of report objectives.

Criteria	Mechanical Stunning	Electrical Stunning	Diathermic Syncope Stunning
**Approval status (Australia)**	Penetrating and non-penetrating permitted for cattle [3,9]	Head-only and head-to-body electrical stunning permitted for cattle [3]	Approved for use under approved arrangements and seeking to be included in current review of standards
**Religious compliance**	Non-penetrating permitted for halal cattle if skull damage is minimal [39] Penetrating may be accepted post-cut [65]	Head-only electric stunning is halal-compliant if parameters set to prevent death [39] Head-to-body is not halal- or kosher-compliant	Halal-compliant [29,39]
**Reliability**	Penetrating captive bolts have 97–98% efficacy, especially when applied in the correct position [75]; non-penetrating captive bolts are slightly less efficacious [75]	Electrical stunning has been associated with between a 1.6% [14] and 10% [57] rate of re-stunning	DTS success rate is 97.8% [30,60,61]
**Reversibility**	Penetrating—irreversible [24,39] Non-penetrating—reversible [24,39]	Head-only—reversible Head-to-body—irreversible [24]	Reversible [62]
**Time to effect**	Instantaneous	Almost instantaneous	1–8 s [18]
**Duration of effect**	Penetrating—permanent effect, i.e., irreversible Non-penetrating—ill-defined, 2–5 min [63]	Some animals may regain consciousness prior to complete exsanguination [25,55,65] Duration of stun: 15–70 s [25]	Energy delivery within the range of 160–220 kJ achieved insensibility of sufficient duration to allow exsanguination using a neck cut alone: 100 -170 s [18]
**Carcase damage**	Penetrating—significant brain tissue damage Non-penetrating—brain tissue damage has been observed	Ecchymoses [84]	No visible damage when energy deliveries of less than 220 kJ are applied [18]
**Logistics**	Safe and secure restraint of the head and body required to ensure accurate placement of the device for effective stunning	Safe and secure restraint of the head and body required to ensure accurate placement of the device for effective stunning	Safe and secure restraint of the head and body required to ensure accurate placement of the device for effective stunning

## Data Availability

No new data were created or analyzed in this study. Data sharing is not applicable to this article

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
