# Peer review of "Commentary: A Comparison of the Methods of the Pre-Slaughter Stunning of Cattle in Australia—Mechanical, Electrical, and Diathermic Syncope"

_animals, 2024, doi:10.3390/ani14213141_

Round 1

Reviewer 1 Report

Comments and Suggestions for Authors

This is a commentary from the Australian point of view comparing the attributes of mechanical, and electromagnetic methods of stunning adult cattle prior to slaughter. Comparisons include Australian approval status, religious compliance, reliability, reversibility, time to effect, duration of effect, carcass damage, cardiovascular effects, logistics, and welfare considerations. A particular emphasis is made comparing the newer technique of reversible stunning made possible by diathermic syncope (DTS), an electromagnetic method for pre-stunning with potential for acceptance by Muslim and possibly Jewish clerical authorities as halal or kosher prior to exsanguination. As noted by the authors, religious non-stun slaughter is a highly issue with relevance to requirements by competent authorities, religious expression, food safety, and animal welfare.

I believe the authors have done a good job summarizing the known issues with current stunning methods and make a compelling argument for more widespread use of the DTS method as a suitable stunning technique for consideration by Muslim and Jewish religious authorities. Specific comments follow:

3.2 Religious compliance: The paradigm of an “essentially contested concept” (ECC) has been applied to a widevariety of deliberative discourse in the social, political, and religious arenas to examine seemingly intractable concepts and generate productive debate. Ul-Ain and Whiting considered animal welfare by rapid exsanguination at slaughter as an ECC; unfortunately, they concluded differing visions of the “good” exist between the competent authorities and religious authorities, such that the method of slaughter is not an ECC where further discourse is likely to result in a negotiated resolution(UlAin Q, Whiting TL. Is a “Good Death” at the Time of Animal Slaughter an Essentially Contested Concept?. Animals. 2017 Dec 14;7(12):99). As to whether DTS will be accepted by religious authorities as kosher, this will likely remain an intractable “essentially contested concept”, as no clear consensus exists thus far (manuscript, lines 212-213).

It would appear that DTS has potential acceptance in the halal market. Fuseini et al, in a survey of UK Islamic scholars and halal consumers, found that most of the scholars (>95%) (CI 86.9 to 98.4%) agreed that if an animal is stunned and then slaughtered by a Muslim and the method of stunning does not result in death, cause physical injury or obstruct bleed-out, the meat would be Halal and 53% (CI 47 to 58%) consumers also thought such meat would be Halal (Fuseini A, Wotton SB, Hadley PJ, Knowles TG. The perception and acceptability of pre-slaughter and post-slaughter stunning for Halal production: The views of UK Islamic scholars and Halal consumers. Meat Science. 2017 Jan 1;123:143-50). This acceptance, however, may depend on local clerical interpretation and may differ with market.

4. Discussion, lines 531-537: The righting reflex is not used as the only indicator of unconsciousness in the slaughter process and several additional reflexes and responses are checked – see Terlouw C, Bourguet C, Deiss V. Consciousness, unconsciousness and death in the context of slaughter. Part II. Evaluation methods. Meat science. 2016 Aug 1;118:147-56; also AVMA Guidelines for Humane Slaughter: 2024 edition (https://www.avma.org/resources-tools/avma-policies/guidelines-humane-slaughter-animals accessed 20 Sept 2024). Most of these indicators are listed in lines 564-573.

5. Conclusion, Animal Welfare Implications, lines 722-725: While DTS might be a suitable stunning technique for consideration by religious authorities, the issue of rapid exsanguination remains an ECC and getting universal acceptance of DTS stunning by religious authorities may be difficult. Acceptance is seemingly more likely by Islamic authorities and halal consumers based on previous published surveys, but may vary with the intended markets.

Author Response

This is a commentary from the Australian point of view comparing the attributes of mechanical, and electromagnetic methods of stunning adult cattle prior to slaughter. Comparisons include Australian approval status, religious compliance, reliability, reversibility, time to effect, duration of effect, carcass damage, cardiovascular effects, logistics, and welfare considerations. A particular emphasis is made comparing the newer technique of reversible stunning made possible by diathermic syncope (DTS), an electromagnetic method for pre-stunning with potential for acceptance by Muslim and possibly Jewish clerical authorities as halal or kosher prior to exsanguination. As noted by the authors, religious non-stun slaughter is a highly issue with relevance to requirements by competent authorities, religious expression, food safety, and animal welfare.

Thank you for this accurate summary of our manuscript.

 I believe the authors have done a good job summarizing the known issues with current stunning methods and make a compelling argument for more widespread use of the DTS method as a suitable stunning technique for consideration by Muslim and Jewish religious authorities. Specific comments follow:

Thank you for these supportive words. 

3.2 Religious compliance: The paradigm of an “essentially contested concept” (ECC) has been applied to a wide variety of deliberative discourse in the social, political, and religious arenas to examine seemingly intractable concepts and generate productive debate. Ul-Ain and Whiting considered animal welfare by rapid exsanguination at slaughter as an ECC; unfortunately, they concluded differing visions of the “good” exist between the competent authorities and religious authorities, such that the method of slaughter is not an ECC where further discourse is likely to result in a negotiated resolution (UlAin Q, Whiting TL. Is a “Good Death” at the Time of Animal Slaughter an Essentially Contested Concept?. Animals. 2017 Dec 14;7(12):99). As to whether DTS will be accepted by religious authorities as kosher, this will likely remain an intractable “essentially contested concept”, as no clear consensus exists thus far (manuscript, lines 212-213).

Thank you for these insights.  We have tried to avoid providing a firm opinion on the religious aspects of DTS and its acceptability by religious authorities.  We hope our current wording reflects our effort to provide scientific evidence.

 It would appear that DTS has potential acceptance in the halal market. Fuseini et al, in a survey of UK Islamic scholars and halal consumers, found that most of the scholars (>95%) (CI 86.9 to 98.4%) agreed that if an animal is stunned and then slaughtered by a Muslim and the method of stunning does not result in death, cause physical injury or obstruct bleed-out, the meat would be Halal and 53% (CI 47 to 58%) consumers also thought such meat would be Halal (Fuseini A, Wotton SB, Hadley PJ, Knowles TG. The perception and acceptability of pre-slaughter and post-slaughter stunning for Halal production: The views of UK Islamic scholars and Halal consumers. Meat Science. 2017 Jan 1;123:143-50). This acceptance, however, may depend on local clerical interpretation and may differ with market.

Thank you again for this further insight into the complexities of navigating religious approval for slaughter methods, including stunning. 

  1. Discussion, lines 531-537: The righting reflex is not used as the only indicator of unconsciousness in the slaughter process and several additional reflexes and responses are checked – see Terlouw C, Bourguet C, Deiss V. Consciousness, unconsciousness and death in the context of slaughter. Part II. Evaluation methods. Meat science. 2016 Aug 1;118:147-56; also AVMA Guidelines for Humane Slaughter: 2024 edition (https://www.avma.org/resources-tools/avma-policies/guidelines-humane-slaughter-animals accessed 20 Sept 2024). Most of these indicators are listed in lines 564-573.

We believe our current wording makes it clear that loss of the righting reflex alone isn’t the only indicator of unconsciousness.  We have also included both the Terlouw reference and the latest AVMA guidelines to substantiate our comments.  Thank you for providing these references.

  1. Conclusion, Animal Welfare Implications, lines 722-725: While DTS might be a suitable stunning technique for consideration by religious authorities, the issue of rapid exsanguination remains an ECC and getting universal acceptance of DTS stunning by religious authorities may be difficult. Acceptance is seemingly more likely by Islamic authorities and halal consumers based on previous published surveys, but may vary with the intended markets.

We agree with your comments wholeheartedly.

Reviewer 2 Report

Comments and Suggestions for Authors

The manuscript contains valuable information, however, I recommend reworking the form in which it is presented to readers. Write separately the methods of conventional defeat stunning, describe perfectly all three methods (mechanical, electrical, dielectric) including the details you have in the discussion. And then focus on cattle stun options for religious slaughter.

Specific comments:

Line 44-45: the expression ….which are all hazards…replace…..which are all risks….

Lines 96-100: The given categorization of stunning methods is inappropriate and should be removed. After all, e.g. exsanguination without stunning cannot be considered a method of stunning! And what is the point of post exsanguination stunning from an animal welfare point of view?!

Lines 169-171: I recommend indicating the time period of the published articles that were included, indicating which databases served as their source, and which keywords were selected to monitor the studied literature.

Line 472: hazards? Rather use risks.

Chapter 4.1.: I recommend describing the different sizes of captive-bolt devices for different categories of slaughtered cattle, including the recommended cartridges for these devices. 

Also, the correct position of placing it on the cattle's head should be described in more detail.

Author Response

The manuscript contains valuable information, however, I recommend reworking the form in which it is presented to readers. Write separately the methods of conventional defeat stunning, describe perfectly all three methods (mechanical, electrical, dielectric) including the details you have in the discussion. And then focus on cattle stun options for religious slaughter.

Thank you for your thoughts.  We hoped this manuscript would serve as a review of stunning methods, with a focus on DTS for comparison to established methods,  to enable consideration of DTS by governments and religious authorities. We have provided numerous references for readers who may want to delve into the details of other methods.

Specific comments:

Line 44-45: the expression ….which are all hazards…replace…..which are all risks….

We have changed the wording as suggested.

Lines 96-100: The given categorization of stunning methods is inappropriate and should be removed. After all, e.g. exsanguination without stunning cannot be considered a method of stunning! And what is the point of post exsanguination stunning from an animal welfare point of view?!

We have modified the wording to ensure that as you rightly say, exsanguination without stunning is not a method of stunning.  We have left post exsanguination stunning in the list however, as although we agree this approach is much less desirable from an animal welfare perspective, it may be better than nothing if stunning prior to exsanguination is not allowed (for whatever reason).

Lines 169-171: I recommend indicating the time period of the published articles that were included, indicating which databases served as their source, and which keywords were selected to monitor the studied literature.

We have added this information as requested.

Line 472: hazards? Rather use risks.

We have changed this wording as requested.

Chapter 4.1.: I recommend describing the different sizes of captive-bolt devices for different categories of slaughtered cattle, including the recommended cartridges for these devices. 

We feel that this information isn’t required but have added a sentence to ensure readers are aware that there are different devices and cartridges available that are suitable for certain scenarios.

Also, the correct position of placing it on the cattle's head should be described in more detail.

We feel that this information isn’t required as mention is already made of the importance of correct placement.  This manuscript isn’t intended to be a manual for users, but a more broad comparison of methods for pre-slaughter stunning.

Reviewer 3 Report

Comments and Suggestions for Authors

The introduction is well-written and reveals the current state of the research field. The purpose of the work and its significance are well-defined. Materials and Methods are described with sufficient detail. The Results and Discussion section provides a concise and precise description of the experimental results, their interpretation, as well as the experimental conclusions that can be drawn. The way the results can be interpreted in the perspective of previous studies is well-made and supported by appropriate bibliography. Overall, I found the article interesting and the subject worthy of being published.
Some additional comments:
Since DTS is a relatively little-known method, it would be advantageous for the reader if the authors described this stunning method in some detail in the Introduction section. In fact, some description of this stunning method only appears in the Discussion section. The authors should also mention the various alternative designations for this method, such as Diathermic Syncope (DTS) and Electromagnetic stunning.
In my opinion, throughout the text, the authors should opt for a single designation (Diathermic Syncope (DTS) or Electromagnetic stunning) rather than using both.
Line 456 – The authors wrote, ‘Electrical stimulation of carcasses is standard practice in beef processing to accelerate the tenderization of meat by rapidly lowering the pH.’ In my view, although electrical stimulation of carcasses generally has the advantage of increasing tenderness, isn’t its use more important to prevent the quality defect known as cold shortening, which results in extremely tough meat that is not tenderizable?
Line 463 – The authors wrote, ‘In Europe, electrical stimulation must not be performed within 30 seconds of thoracic sticking or neck cutting,’ citing the Directive E. Council Regulation (EC) No 1099/2009 on the protection of animals at the time of killing. In: Council E, editor, 2009.
Are the authors sure that this directive states this? This directive, consolidated in 2019, says on page 32:
3. Bleeding of animals
3.2. In the case of simple stunning or slaughter in accordance with Article 4(4), the two carotid arteries or the vessels from which they arise shall be systematically severed. Electrical stimulation shall only be performed once the unconsciousness of the animal has been verified. Further dressing or scalding shall only be performed once the absence of signs of life in the animal has been verified. Please confirm this statement.

Author Response

The introduction is well-written and reveals the current state of the research field. The purpose of the work and its significance are well-defined. Materials and Methods are described with sufficient detail. The Results and Discussion section provides a concise and precise description of the experimental results, their interpretation, as well as the experimental conclusions that can be drawn. The way the results can be interpreted in the perspective of previous studies is well-made and supported by appropriate bibliography. Overall, I found the article interesting and the subject worthy of being published.
Some additional comments:
Since DTS is a relatively little-known method, it would be advantageous for the reader if the authors described this stunning method in some detail in the Introduction section.

We have added a reference in the introduction but feel that a detailed explanation of DTS in the introduction, without equivalent information about other methods of stunning may be unbalanced.  The text in the discussion serves to ensure the reader understands the main differences between the methods.

In fact, some description of this stunning method only appears in the Discussion section. The authors should also mention the various alternative designations for this method, such as Diathermic Syncope (DTS) and Electromagnetic stunning.

We only want to talk about DTS as other methods of electromagnetic stunning are not used in Australia.  We have clarified this in a number of places in the text to ensure the reader understands that we are not discussing other electromagnetic methods.

In my opinion, throughout the text, the authors should opt for a single designation (Diathermic Syncope (DTS) or Electromagnetic stunning) rather than using both.

We have adjusted the text to use the term DTS consistently.

Line 456 – The authors wrote, ‘Electrical stimulation of carcasses is standard practice in beef processing to accelerate the tenderization of meat by rapidly lowering the pH.’ In my view, although electrical stimulation of carcasses generally has the advantage of increasing tenderness, isn’t its use more important to prevent the quality defect known as cold shortening, which results in extremely tough meat that is not tenderizable?

We have adjusted the wording to acknowledge your point.

Line 463 – The authors wrote, ‘In Europe, electrical stimulation must not be performed within 30 seconds of thoracic sticking or neck cutting,’ citing the Directive E. Council Regulation (EC) No 1099/2009 on the protection of animals at the time of killing. In: Council E, editor, 2009.
Are the authors sure that this directive states this? This directive, consolidated in 2019, says on page 32:

  1. Bleeding of animals
    3.2. In the case of simple stunning or slaughter in accordance with Article 4(4), the two carotid arteries or the vessels from which they arise shall be systematically severed. Electrical stimulation shall only be performed once the unconsciousness of the animal has been verified. Further dressing or scalding shall only be performed once the absence of signs of life in the animal has been verified. Please confirm this statement.

We have corrected this statement – your comments are correct, and we apologise for this issue.

Reviewer 4 Report

Comments and Suggestions for Authors

This review describes stunning methods for cattle pre-slaughter. The focus is on reversable methods and a comparison with a novel method. There is a lack of research in this area and a review of the available research will be a great resource. However, this review is a little confusing in places and not quite complete. Some statements are not quite accurate. The search terms may not have found all relevant research, I have added a few extra references that I found at the bottom of my comments. It would be good to introduce the different stunning methods in the introduction, particularly regarding the novel method (DTS). Some other details in the introduction could probably be removed to concentrate on factors directly affected by stunning method. I also wonder if it would be better to concentrate this review on just the reversable methods of stunning, to achieve a clearer comparison.

Specific comments:

L120, exsanguination preferred as cause of death for meat quality and food safety not quite true. Animals can, and are, killed by some stunning methods without affecting meat quality of food safety.

L125 While the duration cited relate to the use of the neck cut only, insensibility is much faster if an additional thoracic stick is used.

L129 Please cite original study (Blackmore, 1984).

Method: It is recommended to limit the date of publication, to recognise improvements in methods over time. Maintenance and correct ammunition (including pneumatics) of captive bolt stunners has much improved in recent years. Similarly, the more modern electrical stunning unit have automatic feedback to ensure enough current is delivered.

L212-213. ‘no clear consensus on whether DTS may be suitable for kosher slaughter’. This suggests that some may consider it suitable. This seems unlikely, based on the general position on stunning and the non-acceptance of electrical stunning.

L235-236. Observations from commercial abattoirs are highly variable and more detail needs to be provided to qualify certain results. For example ‘29% of cattle and 12% of cattle respectively required two or more shots to induce unconsciousness is not quite true. These animals received a second shot (which may just be a ‘security shot’, when the aim was not ideal). The same paper also states that 91% (NPCB) and 99% (PCB) collapsed on the first shot. This study involves Zebu cattle, which may be less suitable for NPC.

L236-238 ‘repeat shots’ in cattle (including bulls, steers and heifers) stunned with a penetrating captive bolt was 4.2% despite 8.7% of these animals being ‘imperfectly stunned’. This reference is not listed in table 1. The study itself discusses the findings and concludes that 7.7% had a shallow depth of unconsciousness (imperfect stun).

L243-244 ‘. Electrical stunning has been associated with a 10% rate of re-stunning’, this turn of phrase is a little strange. It is also used in Table 2, despite more reliable data available from an Australian study (Hewitt, 2016).

L316-317 ‘ it may take 55 seconds for death by exsanguination’. Clarify that this is when both carotid arteries and both jugular veins are severed. Often a thoracic stick is used in addition, to speed up blood loss. According to reference 52, time to insensibility would be 30s.

L435 ‘DTS requires a rotating box’, not mentioned before? Can this be clarified?

L461-466 This is confusing. Electro-immobilisation is allowed after the animal is insensible (30s after sticking), see ref 52. This does not appear to apply to NPCB (although it is being used in Australian abattoirs, so is this used against the direction?).

Section 3.9.5 talks about safety and mentions the use of electro-immobilisers. If sticking is performed during the tonic phase, a delay of 30 seconds is not necessarily a problem for worker safety (but may be important if cattle is shackled in that time). This should be clarified.

Section 3.10 does not really contribute anything. Stunning methods should be instantaneous, or should not induce cause pain or fear during induction. It is not clear what the impact is of DTS, as the paper is not yet published.

The discussion does not provide an in-depth evaluation of the literature described in the results. Some of the descriptions of the different methods of stunning are better placed in the introduction. A clear comparison of methods, and limitations of the available research, should form part of the discussion. This should include a comment on Zebu type cattle, and the suitability of the different stunning methods. It would also be useful to comment on the use of the “Malaysia’ protocol, for which DTS may be more suitable than the current stunning methods.

Additional references:

Lücking, A., Louton, H., von Wenzlawowicz, M., Erhard, M., & von Holleben, K. (2024). Movements after Captive Bolt Stunning in Cattle and Possible Animal-and Process-Related Impact Factors—A Field Study. Animals14(7), 1112.

Casagrande, R. R., Alexander, L., & Edwards-Callaway, L. N. (2020). Effects of penetrating captive bolt gun model and number of stuns on stunning-related variables of cattle in a commercial slaughter facility. Meat Science, 170, 108231.

Setijanto, H., Nuraini, H., Nisa, C., Novelina, S., Cahyadi, D. D., Sudarnika, E., & Agungpriyono, S. (2024). The Success Rate of Non-Penetrative Pre-Slaughter Stunning on Australian Brahman Cross Cattle Slaughter in Indonesia. Acta Vet Indones. The Indonesian Veterinary Journal/Jurnal Acta Veterinaria Indonesiana12(1).

Author Response

This review describes stunning methods for cattle pre-slaughter. The focus is on reversable methods and a comparison with a novel method. There is a lack of research in this area and a review of the available research will be a great resource. However, this review is a little confusing in places and not quite complete. Some statements are not quite accurate. The search terms may not have found all relevant research, I have added a few extra references that I found at the bottom of my comments. It would be good to introduce the different stunning methods in the introduction, particularly regarding the novel method (DTS). Some other details in the introduction could probably be removed to concentrate on factors directly affected by stunning method. I also wonder if it would be better to concentrate this review on just the reversable methods of stunning, to achieve a clearer comparison.

We wanted to compare methods that are currently approved in Australia and feel that the current inclusions are suitable. While we appreciate your thoughts about concentrating only on reversible methods we would lose the value of acknowledging the limitations of the few methods that are actually approved here.

Moving the details about the different methods to the introduction would make the introduction very long, which is why we have left them in the discussion.  The introduction serves to provide a background to the considerations for the impact of different methods of pre-slaughter stunning so the reader has a grasp of the nuances of different methods.

Specific comments:

L120, exsanguination preferred as cause of death for meat quality and food safety not quite true. Animals can, and are, killed by some stunning methods without affecting meat quality of food safety.

We appreciate your comment but feel that exsanguination in accordance with both Islamic and Jewish criteria is important for both food safety and meat quality as well as cultural and religious reasons.

L125 While the duration cited relate to the use of the neck cut only, insensibility is much faster if an additional thoracic stick is used.

Thank you for this insight. 

L129 Please cite original study (Blackmore, 1984).

The original study has been cited as suggested.

Method: It is recommended to limit the date of publication, to recognise improvements in methods over time. Maintenance and correct ammunition (including pneumatics) of captive bolt stunners has much improved in recent years. Similarly, the more modern electrical stunning unit have automatic feedback to ensure enough current is delivered.

Thank you for those comments – we agree. 

L212-213. ‘no clear consensus on whether DTS may be suitable for kosher slaughter’. This suggests that some may consider it suitable. This seems unlikely, based on the general position on stunning and the non-acceptance of electrical stunning.

We have not got a clear understanding on whether DTS is either accepted or rejected by Jewish authorities, therefore we prefer to remain open with the statement  ‘no consensus’.

L235-236. Observations from commercial abattoirs are highly variable and more detail needs to be provided to qualify certain results. For example ‘29% of cattle and 12% of cattle respectively required two or more shots to induce unconsciousness is not quite true. These animals received a second shot (which may just be a ‘security shot’, when the aim was not ideal). The same paper also states that 91% (NPCB) and 99% (PCB) collapsed on the first shot. This study involves Zebu cattle, which may be less suitable for NPC.

Thank you for these comments.  We are not saying that these data are ideal, but they do remain however, the best data that exists.  We have not critiqued such data in this review as our aim is to highlight the attributes of DTS compared to other methods, based on published data, albeit imperfect.

L236-238 ‘repeat shots’ in cattle (including bulls, steers and heifers) stunned with a penetrating captive bolt was 4.2% despite 8.7% of these animals being ‘imperfectly stunned’. This reference is not listed in table 1. The study itself discusses the findings and concludes that 7.7% had a shallow depth of unconsciousness (imperfect stun).

Table 1 refers only to non-penetrative captive bolts – as there were multiple references to include. This Gregory reference is about penetrating captive bolts so was written in the text and not included in the table.

L243-244 ‘. Electrical stunning has been associated with a 10% rate of re-stunning’, this turn of phrase is a little strange. It is also used in Table 2, despite more reliable data available from an Australian study (Hewitt, 2016).

The phrase ‘rate of re-stunning’ is the expression used by the authors.  Both studies are now referenced in table 2.

L316-317 ‘ it may take 55 seconds for death by exsanguination’. Clarify that this is when both carotid arteries and both jugular veins are severed. Often a thoracic stick is used in addition, to speed up blood loss. According to reference 52, time to insensibility would be 30s.

This detail has been clarified to include a comment that the figure relates to when both carotid arteries and both jugular veins are severed.

L435 ‘DTS requires a rotating box’, not mentioned before? Can this be clarified?

This has been clarified. A standing crush can also be used.

L461-466 This is confusing. Electro-immobilisation is allowed after the animal is insensible (30s after sticking), see ref 52. This does not appear to apply to NPCB (although it is being used in Australian abattoirs, so is this used against the direction?).

The Australian requirements state: Electro-immobilization after reversible stunning and prior to sticking is not acceptable.

This practice is not acceptable where an animal has been reversibly stunned and does not comply with the NCCAW position on electro-immobilization (m). The practice allows for the perpetuation of poor stunning practices through masking deficiencies in the area of effective stunning technique.

AQIS will only accept the practice in the following situations where

  1. a captive penetrating bolt stun has been used effectively on cattle as determined

by the Standard Operating Procedure – Animal Care – Penetrating Captive Bolt

Stunning and Sticking of Cattle,

  1. reversible electrical stunning is used provided the animal has reached irreversible insensibility after sticking as determined by the guidelines identified on page 10 of these notes.

Text has been added to clarify.

Section 3.9.5 talks about safety and mentions the use of electro-immobilisers. If sticking is performed during the tonic phase, a delay of 30 seconds is not necessarily a problem for worker safety (but may be important if cattle is shackled in that time). This should be clarified.

Thank you for the comment but we feel it relates more to operational considerations, which aren’t the focus of this review.

Section 3.10 does not really contribute anything. Stunning methods should be instantaneous, or should not induce cause pain or fear during induction. It is not clear what the impact is of DTS, as the paper is not yet published.

We feel that this section is important and should be included.  As more evidence emerges a greater understanding of welfare impacts will evolve.

The discussion does not provide an in-depth evaluation of the literature described in the results. Some of the descriptions of the different methods of stunning are better placed in the introduction. A clear comparison of methods, and limitations of the available research, should form part of the discussion. This should include a comment on Zebu type cattle, and the suitability of the different stunning methods. It would also be useful to comment on the use of the “Malaysia’ protocol, for which DTS may be more suitable than the current stunning methods.

We have considered moving text around to have details about the methods of stunning in the introduction, but as other reviewers have been satisfied with the structure, we have left it as is. We believe the introduction serves to highlight the factors that impact upon the suitability of different techniques e.g., related to meat quality etc.

We acknowledge that the best way to compare the methods of pre-slaughter stunning is a prospective trial using a variety of animal types and methods of stunning.  This kind of study has not been performed with DTS and for the purposes of this manuscript we can only present the evidence as it stands.

Additional references:

Lücking, A., Louton, H., von Wenzlawowicz, M., Erhard, M., & von Holleben, K. (2024). Movements after Captive Bolt Stunning in Cattle and Possible Animal-and Process-Related Impact Factors—A Field Study. Animals14(7), 1112.

Casagrande, R. R., Alexander, L., & Edwards-Callaway, L. N. (2020). Effects of penetrating captive bolt gun model and number of stuns on stunning-related variables of cattle in a commercial slaughter facility. Meat Science, 170, 108231.

Setijanto, H., Nuraini, H., Nisa, C., Novelina, S., Cahyadi, D. D., Sudarnika, E., & Agungpriyono, S. (2024). The Success Rate of Non-Penetrative Pre-Slaughter Stunning on Australian Brahman Cross Cattle Slaughter in Indonesia. Acta Vet Indones. The Indonesian Veterinary Journal/Jurnal Acta Veterinaria Indonesiana12(1).

 Thank you for these references. We have included them throughout the text

Round 2

Reviewer 2 Report

Comments and Suggestions for Authors

Chapter 2 Materials and Methods lacks details of the databases the authors drew on. I recommend adding more.

Author Response

Comment 1:

Materials and Methods lacks details of the databases the authors drew on. I recommend adding more.

More detail is included about the review methodology.

Reviewer 4 Report

Comments and Suggestions for Authors

While some changes have been made, it still seems that the review lacks clear focus and contains some inconsistencies. I have added my comments to the attached document.
